# Introduction of the difluoromethyl group at the meta- or para-position of pyridines through regioselectivity switch

Pengwei Xu [1,2], Zhe Wang[1,2], Shu-Min Guo[1] & Armido Studer [1] ✉

Difluoromethyl pyridines have gained significant attention in medicinal and agricultural chemistry. The direct C−H-difluoromethylation of pyridines represents a highly efficient economic way to access these azines. However, the direct meta-difluoromethylation of pyridines has remained elusive and methods for site-switchable regioselective meta- and para-difluoromethylation are unknown. Here, we demonstrate the meta-C−H-difluoromethylation of pyridines through a radical process by using oxazino pyridine intermediates, which are easily accessed from pyridines. The selectivity can be readily switched to para by in situ transformation of the oxazino pyridines to pyridinium salts upon acid treatment. The preparation of various meta- and para-difluoromethylated pyridines through this approach is presented. The mild conditions used also allow for the late-stage meta- or para-difluoromethylation of pyridine containing drugs. Sequential double functionalization of pyridines is presented, which further underlines the value of this work.

The incorporation of fluorinated moieties into the framework of bioactive compounds is of high importance in medicinal and agricultural chemistry, as fluorinated entities can modulate the biological and physiological activity of a compound by enhancing its lipophilicity, bioavailability, and metabolic stability[1–5]. Among these fluorinated moieties, the difluoromethyl group (CF$_2$H) occupies a special role, as it possesses an acidic proton, which may interact with the targeting enzymes through hydrogen bonding and accordingly CF$_2$H can serve as a bioisostere of alcohol, thiol, and amine moieties[6]. Given the prevalence of pyridines in drugs and agrochemicals, such an incorporation strategy has been successfully utilized in marketed difluoromethyl pyridines[7–9] (Fig. 1a). In light of this, the development of methods for rapid access to diverse difluoromethylated pyridines will most likely facilitate drug discovery and may lead to the discovery of relevant novel candidates.

Simple difluoromethyl pyridines can be prepared from acyclic precursors. However, this strategy represents an inefficient way, and only a low variability in candidate structures can be addressed by de novo synthesis. In discovery campaigns, the preferable way is to directly introduce difluoromethyl moieties into existing pyridines. In this context, metal-catalyzed cross-couplings[10–16] can partly realize this demand, while C−H-difluoromethylation of pyridines without pre-installed functional groups is the ideal approach, as it can substantially increase step economy[17–21] (Fig. 1b). To date, the ortho C−H-difluoromethylation of pyridines has been well studied[22–30]. As an example, Baran[22,23] applied Zn(SO$_2$CF$_2$H)$_2$ as the CF$_2$H-radical source in Minisci-type chemistry. In contrast, regioselective pyridine para-difluoromethylation had not been explored until recently when McNally and coworkers reported a successful para-functionalization using phosphonium salts[31]. However, the meta-C−H-difluoromethylation of pyridines remains an unsolved challenge to date[17]. In addition, no methods have been disclosed for the site-switchable C−H-difluoromethylation of pyridines under easily tunable conditions [18–21].

Known methods for C−H-functionalization of pyridines are largely restricted to the ortho- and para-positions due to the electronic nature of the pyridine core[17–21,32]. Despite great achievements in the area of transition-metal catalysis[33–40], temporary dearomatization approaches

[1]Organisch-Chemisches Institut, Universität Münster, Corrensstrasse 40, 48149 Münster, Germany. [2]These authors contributed equally: Pengwei Xu, Zhe Wang. ✉e-mail: studer@uni-muenster.de

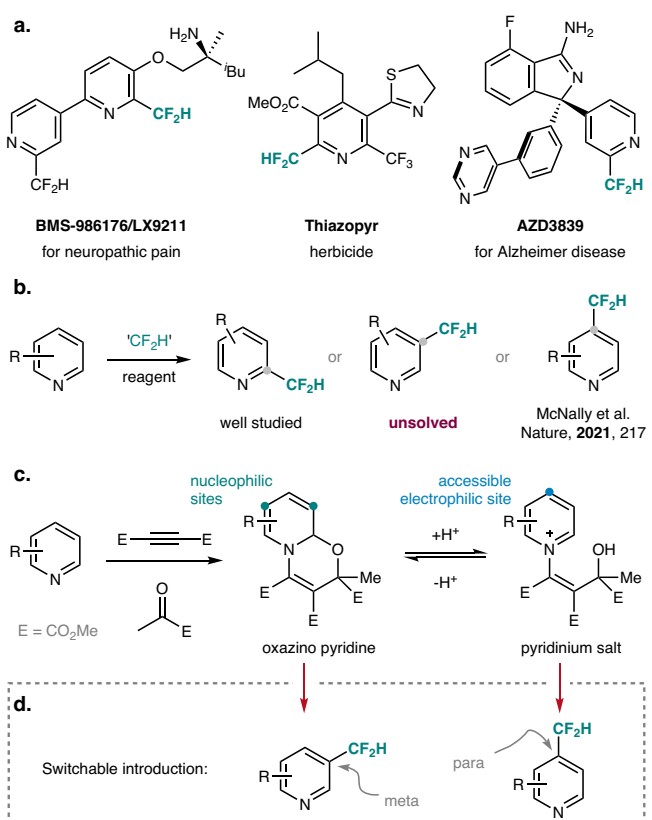

**Fig. 1 | Site-selective C−H difluoromethylation of pyridines. a** Bioactive compounds containing a difluoromethylated pyridine moiety. **b** Three possible regioisomers for the C−H difluoromethylation of pyridines. **c** Our design: switchable C−H functionalization of pyridines. Oxazino pyridines that are present under basic conditions show nucleophilic reactivity at the β- and δ-positions, while the pyridinium ions formed under acidic conditions show electrophilic reactivity at the γ-position. **d** The two difluoromethylated pyridines that can be regioselectively accessed upon switching from oxazino pyridines to pyridinium ions.

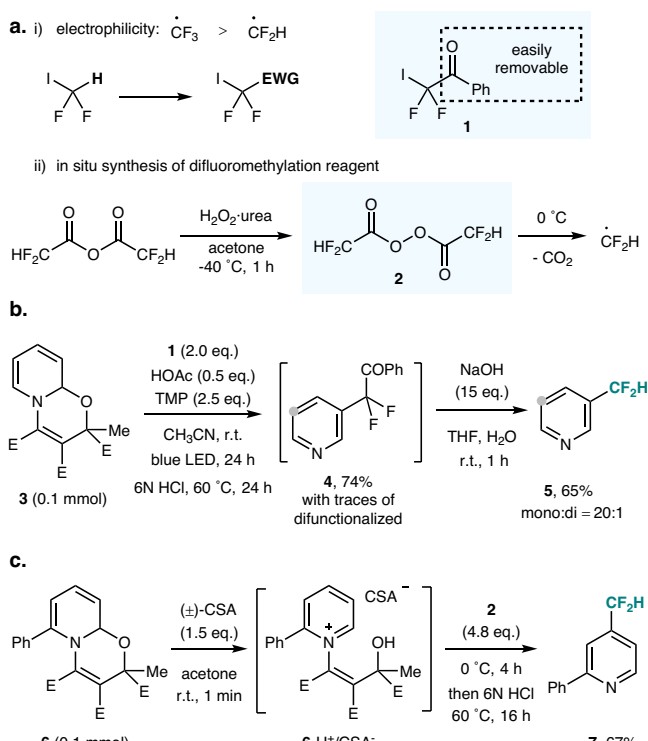

**Fig. 2 | Establishment of *meta* and *para* C−H difluoromethylation of pyridines. a** Choice of difluoromethylation reagent. **b** *meta*-functionalization. **c** *para*-functionalization. Yield of **5** was determined by $^{19}$F NMR using 1-bromo-4-fluorobenzene as the internal standard. E = CO$_2$Me. EWG electron-withdrawing group, TMP 2,2,6,6-tetramethylpiperidine, CSA camphorsulfonic acid.

## Results and discussion

We initially tested CF$_2$IH as the difluoromethyl radical source to realize oxazino pyridine difluoromethylation. However, the electrophilicity of the difluoromethyl radical (•CF$_2$H) is lower than that of the trifluoromethyl radical (•CF$_3$) (Fig. 2a−i)[55] and difluoromethylation of the oxazino pyridine **3** could not be achieved under the tested conditions. Indeed, the difluoromethyl radical is considered as a nucleophilic radical[22,56,57]. Therefore, the difluoromethyl reagent has to be equipped with a removable electron-withdrawing auxiliary group, which will enhance the electrophilicity of the corresponding difluorinated C-radical (•CF$_2$H to [•CF$_2$(EWG)])[55,58]. The auxiliary EWG should subsequently be removed through protodefunctionalization, and we selected the 2,2-difluoro-2-iodo-1-phenylethan-1-one (**1**) as reagent, since it is easily accessible and known as a C-radical precursor[59–62]. We commenced our investigations with the difluoromethylation of the oxazino pyridine **3** and after extensive experimentation we found that the reaction is best conducted in acetonitrile in the presence of acetic acid (0.5 equiv.) and 2,2,6,6-tetramethylpiperidine (TMP, 2.5 equiv.) upon irradiation (LED, 456 nm) for 24 h at room temperature (Fig. 2b). Subsequent addition of 6 N HCl (one-pot procedure) gave the meta-difluorobenzoylmethylated pyridine **4** in 74% yield containing traces of the meta,meta'-difunctionalized congener (for details on reaction optimization, see Supplementary Fig. 3). Subsequently, the benzoyl group could be readily removed upon addition of NaOH (same pot) and the targeted difluoromethylated pyridine **5** was obtained in 65% yield containing a small amount of the corresponding bisdi-fluoromethylated pyridine (mono:di = 20:1).

Addressing the para-difluoromethylation through the reaction of pyridinium salts, bis(difluoroacetyl) peroxide (**2**) was selected as the •CF$_2$H source, which could be easily generated in situ from commercial difluoroacetic anhydride and urea·H$_2$O$_2$ at −40 °C. Homolyis of **2** occurs at 0 °C to generate the difluoromethyl radical along with CO$_2$

have emerged as highly promising tools for the meta-functionalization of pyridines. Applying such strategies, electron-deficient pyridines are first transformed into activated electron-rich intermediates, which can then undergo electrophilic reactions followed by rearomatization to finally provide meta-substituted pyridines. Along these lines, the Wang group[41–45] and others[46–50] demonstrated reductive dearomatization, electrophile-trapping, and oxidation sequences through dihydropyridine intermediates. The McNally group developed a meta-functionalization via a ring opening, halogenation, and ring-closing sequence through Zincke imines[51]. Our group introduced a redox-neutral dearomatization–rearomatization process for the versatile meta-functionalization of pyridines, where azines are first dearomatized with acetylenedicarboxylate (DMAD) and methyl pyruvate (MP) to afford bench-stable oxazino pyridine intermediates in excellent yields (Fig. 1c)[52]. These can undergo site-selective reactions via radical or ionic pathways, and rearomatization then leads to the meta-functionalized pyridines. In addition, para-selective functionalization by protonation of the oxazino pyridines and subsequent Minisci-type radical alkylation of the corresponding pyridinium salts is feasible[53,54]. By taking advantage of these two strategies, we herein report a meta- and site-switchable meta- and para-C−H-difluoromethylation of pyridines with easily available CF$_2$H-radical sources (Fig. 1d). This site-switchable difluoromethylation is also applicable for the late-stage modification of pyridine-containing drugs. The effectiveness and practicability are further featured by one-pot procedures and sequential para, meta-functionalizations, which allow access to diverse difluoro-methylated pyridines.

**Fig. 3 | Substrate scope.** [a]Yields were based on isolated oxazino pyridines. [b]Yield was determined by ¹⁹F NMR using 1-bromo-4-fluorobenzene as the internal standard. DMAD: dimethyl acetylenedicarboxylate. MP: methyl pyruvate. m:d: ratio of mono:di at the meta position. r.r. regioselectivity in the oxazino pyridine (δ:β-selectivity).

(Fig. 2a–ii)[56,58]. To our knowledge, this readily available radical difluoromethylation reagent studied by Sodeoka for alkene difluoromethylation[58] has not yet been used in Minisci-type alkylations. Reaction optimization was conducted on oxazino pyridine **6**. Treatment of **6** with (±)-camphorsulfonic acid (CSA, 1.5 equiv.) in acetone leads to the corresponding pyridinium salt **6**-H⁺/CSA⁻ that was reacted with **2** (4.8 equiv.) at 0 °C for 4 h. After addition of 6 N HCl in one-pot and subsequent heating to 60 °C (16 h), the para-difluoromethylated product was obtained in 67% yield with complete regioselectivity (Fig. 2c). The utilization of a low temperature during radical difluoromethylation

contributes to achieving a high degree of para-selectivity. It is important to highlight that both of our methods do not require any transition metal, and reagents used are either commercial or readily prepared. For further insight into the potential mechanisms underlying both processes, we refer to the Supplementary Information.

Under optimized conditions a range of differently substituted pyridines with varied electronic properties could be regioselectively difluoromethylated at the meta- as well as the para-position (Fig. 3). Considering the meta-functionalization, substrates bearing two free C−H meta-positions (**5, 8-17**), including the parent pyridine mostly provided

the mono-difluoromethylated product along with the corresponding meta,meta′-bisdifluoromethylated pyridine as a minor byproduct. The initial radical difluoromethylation always preferred to occur at the more reactive δ-position of the dienamine entity on the oxazino pyridine (**14-17**), as a more stabilized radical intermediate is formed through δ-addition, which is governed by a larger resonance stabilization. This high δ-regioselectivity was previously also observed for the radical meta-trifluoromethylation[52]. In nearly all cases, the mono-functionalized product was formed with complete δ-regioselectivity. Only for oxazino pyridines that carry electron-rich aryl groups on the α-position, the β-regioisomer was identified as a minor byproduct (see **14-16**). Considering double difluoromethylation, once the first $CF_2COPh$-group is installed, the remaining β-position is less nucleophilic, because of the electron-withdrawing effect exerted by the $CF_2COPh$-group. Accordingly, selective mono-difluoromethylation was achieved in several cases (ratio m:d > 20:1, **9, 12, 13, 15, 17** and **26**). However, the β-position of the δ-$CF_2COPh$-functionalized oxazino pyridine remains reactive depending on the additional oxazino pyridine substituents. Considering γ-arylated oxazino pyridines as substrates, mono/di-functionalization selectivity decreases as a function of the electron-donating ability of the para-substituent on the aryl group, in line with our hypothesis (see **8, 9** and **10**). Further, for α- and γ-substituted oxazino pyridines, steric effects will likely also slow down the second C−H-functionalization. The excellent mono-functionalization selectivity observed for the γ-phenoxy oxazino pyridine is currently not understood (see **13**). In general, pyridines bearing meta-substituents (**18-21**) provided slightly lower yields for the meta-functionalization. This is likely due to the fact that in these cases, the oxazino pyridines were formed as regioisomeric mixtures, and consequently, the intrinsically more reactive δ-position of one regioisomer was blocked, contributing to the reduced yields. The relatively moderate yields observed in meta-difluoromethylation are, to some extent, attributed to product loss during the process of isolation and purification (refer to Supplementary Fig. 7).

Disubstituted pyridines with substituents on 2,3-, 2,4- or 2,5-position were also investigated (**22-32**, 45−78%). The 2-position could be decorated with aryl, heteroaryl and alkynyl groups, and the 3-, 4-, 5-positions could be aryl, alkyl, halo, and methoxy substituted. Of note, the electron-rich thiophene moiety was not difluoromethylated (**25, 28, 30, 31**, 49−78%), showing that our dearomatization strategy allows us to functionalize intrinsically far less reactive pyridines in the presence of more reactive heteroarenes. With 2,4-diphenyl pyridine, mono-difluoromethylation was achieved exclusively and the meta,-meta′-di-functionalization did not occur, likely for steric reasons (**26**, 55%). The π-donor property of 2-aryl and 2-alkynyl groups in 2,5-disubstituted pyridines increased the reactivity at the β-position of the oxazino pyridine and difluoromethylation occurred in good yields (**27-32**, 48−78%). Thienopyridine and furopyridine were eligible substrates to afford **33** (64%) and **34** (52%). The intrinsically more reactive thieno- and furo-entities in these interesting heteroarenes remained untouched. Quinolines also worked; however, reactions were less efficient, and unfunctionalized quinolines could be recovered (**35, 36**). Isoquinolines could not be difluoromethylated through this strategy.

We next investigated the scope of the para-functionalization. 2-Arylated pyridines bearing electron-rich and also electron-poor substituents at different positions reacted highly regioselectively to the corresponding para-difluoromethylated pyridines **7** and **37-40** in 51−73% yield. The 4-(difluoromethyl)pyridine **37** lacking any additional substituent was obtained in 61% along with 15% of the corresponding ortho,para-bistrifluoromethylated pyridine. Of note, ortho-difluoromethylation was not observed for all other cases, showing the very good ortho-shielding effect of the N-substituent in these pyridinium salts. The amide NH-group was tolerated, as shown by the successful preparation of **41** (38%). 2,3- and 2,5-disubstituted pyridines were eligible substrates (**42-45**), and even a 2,3,5-trisubstituted pyridine could be para-difluoromethylated to afford the highly

substituted pyridine **47** in 47% yield. Difluoromethylation of furopyridine was achieved with complete para-selectivity in 68% yield, and the furan core did not react (**47**).

## Synthetic applications

We were pleased to find that our two methods are also applicable to the regioselective meta or para-difluoromethylation of pyridine-containing drugs and drug derivatives (Fig. 4a). For example, loratadine was regioselectively difluoromethylated either at the meta or at the para-position of its pyridine moiety in 60% and 57% yield, respectively (**48,49**). An ibuprofen derivative and nicotinyl alcohol were successfully meta-functionalized (**50,51**), albeit in moderate yields, and the unfunctionalized starting pyridines could be partially recovered. Moreover, $CF_2H$ groups were successfully introduced to the pyridine para-position of nikethamide (**52**) and vismodegib (**53**). All reactions displayed in Figs. 3 and 4a were conducted as two-pot procedures with oxazino pyridines used as isolated starting materials for the subsequent C−H functionalization step. To further improve the practicability of our methods, one-pot reactions without isolating the dearomatized intermediates were performed. Along these lines, one-pot meta- and para-difluoromethylation of loratadine at a larger scale was realized (Fig. 4b). Comparing with the two-pot reactions shown in Fig. 4a, yields were only slightly decreased, demonstrating the potential of these methods in process chemistry. Finally, consecutive regioselective ionic and radical double functionalization of the C−H bonds in pyridines was studied on the ligand-derived oxazino pyridine **54** (Fig. 4c). The sequence commenced with the ionic meta-chlorination of **54** in $CH_2Cl_2$ with N-chlorosuccinimide[52]. After removal of the solvent, the residue was subjected to our radical difluoromethylation procedures in the same flask, and the 3,5- or 3,4-difunctionalized pyridines **55** and **56** were isolated in 38% and 32% overall yield. In addition, the protected difluoromethyl group was first introduced at the meta position of the oxazino pyridine **54**. The intermediate was isolated, and the installation of a para-$CF_2H$ moiety proceeded smoothly. Rearomatization and deprotection finally delivered the meta,para-bisdifluoromethylated pyridine **57**, which would be difficult to prepare by other methods, in 31% overall yield.

In summary, we have realized a meta and site-switchable meta- and para-C−H-difluoromethylation of pyridines. These C−H-functionalizations proceed through a redox-neutral dearomatization-rearomatization sequence with oxazino pyridine intermediates as the substrates for the radical meta-C−H-difluoromethylation. The para-C−H difluoromethylation was accomplished by in situ transformation of the oxazino pyridines to the corresponding pyridinium salts upon acid treatment and subsequent highly regioselective Minisci-type alkylation. Due to the easy availability of the two difluoromethylation reagents, mild reaction conditions, easy operations, broad scope, late-stage applications, and especially the site-switchablility, these practical methods should find use in pharmaceutical and agrochemical industry.

## Methods

### General procedure A for meta-difluoromethylation

To an oven-dried 10 mL Schlenk tube, an oxazino pyridine (0.20 mmol, 100 mol%) was added. The tube was capped and evacuated/refilled with argon for three times. Under an argon flow, $CH_3CN$ (2 mL), 2,2,6,6-tetramethylpiperidine (TMP, 0.50 mmol, 85 μL, 250 mol%), HOAc (0.10 mmol, 5.6 μL, 50 mol%) and 2,2-difluoro-2-iodo-1-phenylethan-1-one **1** (0.40 mmol, 64 μL, 200 mol%) were sequentially added via syringe. The tube was capped again and placed in a photoreactor, stirred, and irradiated for 24 h. The temperature was maintained below 30 °C using a fan. Afterwards, 6 N HCl (2 mL) was added to the reaction mixture, and the tube was heated at 60 °C for 24 h. The reaction mixture was basified with saturated aqueous $Na_2CO_3$ solution (30 mL) and extracted with EtOAc (10 mL × 3). The combined organic phase was dried over $Na_2SO_4$, filtered, and concentrated under vacuum. The residue was then

**Fig. 4 | Synthetic applications and difunctionalizations. a** Late-stage difluoromethylation of drugs and drug derivatives. **b** One-pot difluoromethylation of loratadine. **c** Consecutive meta,meta'- or meta,para-difunctionalization of pyridines. NCS N-chlorosuccinimide.

dissolved with THF and treated with water (100 μL) and NaOH (3 mmol, 120 mg, 15 equiv.). After stirring at r.t. for 1 h, the reaction mixture was diluted with brine (30 mL) and extracted with Et₂O or EtOAc (10 mL × 3). The combined organic phase was dried over Na₂SO₄, filtered, concentrated, and submitted to flash column chromatography (pentane/EtOAc) to yield the meta-difluoromethylated pyridine.

### General procedure B for para-difluoromethylation

Under argon, difluoroacetic anhydride (0.50 mL, 4.0 mmol, 20 equiv.) was slowly added to a suspension of urea·H₂O₂ (90.4 mg, 0.96 mmol, 4.8 equiv.) in dry acetone (1.0 mL) in a 10 mL Schlenk tube equipped with a magnetic stirring bar at −40 °C, and the mixture was stirred for 1 h at the same temperature. Another 10 mL oven-dried Schlenk tube equipped with a magnetic stirring bar was charged with oxazino pyridine (0.2 mmol, 1.0 equiv.) and (±)-camphorsulfonic acid ((±)-CSA, 69.7 mg, 0.3 mmol, 1.5 equiv.) and subjected to three cycles of vacuum/argon backfill. Then, dry acetone (1.0 mL) was added and the reaction mixture was cooled down to 0 °C using an ice/water bath. Afterwards, the former reaction mixture at −40 °C was transferred to the later tube at the ice/water bath using a glass pipette under an argon flow. The reaction mixture was stirred for 4 h at 0 °C. After completion, 6 N HCl (4 mL) was added to the reaction mixture, and the tube was heated at 60 °C for 16 h. The mixture was basified with saturated aqueous Na₂CO₃ solution and extracted with EtOAc (10 mL × 3). The combined organic layer was dried (over Na₂SO₄), filtered, and concentrated under reduced pressure. The residue was subjected to flash

column chromatography over silica gel to give the corresponding product.

## Data availability

Supplementary information and chemical compound information accompany this paper at www.nature.com/ncomms. The data supporting the results of this work are included in this paper or in the Supplementary Information and are also available upon request from the corresponding author.

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

## Acknowledgements

We thank the Deutsche Forschungsgemeinschaft (DFG), the Alexander von Humboldt Foundation (post-doctoral fellowship to P.X.), and the China Scholarship Council (PhD fellowship to Z.W.) for supporting this work. We also thank Dr. K. Bergander, University of Münster, for conducting NMR experiments and Dr. M. Letzel, University of Münster, for MS analysis.

## Author contributions

A.S. and P.X. conceived and designed the project. P.X. and Z.W. performed the experiments and analyzed the data. P.X. Z.W., S.-M.G., and A.S. contributed to the discussion and wrote the manuscript.

## Funding

## Competing interests

The authors declare no competing interest.
