## [Peer Review File · Nature Communications]

Introduction of the difluoromethyl group at the meta- or para-position of pyridines through regioselectivity switchREVIEWER COMMENTS

Reviewer #1 (Remarks to the Author):

The authors report the introduction of difluoromethyl group into the meta- and para-positions of pyridine derivatives. The meta-selectivity is achieved by the radical addition of difluoromethyl radical to dearomatized pyridine derivatives by ozazino pyridine intermediates. On the other hand, the para-selectivity is realized by in situ transformation of the oxazino pyridines to pyridinium salts. The authors also investigated late-stage meta- and para-difluoromethylation of pyridine derivatives including drugs and double functionalization of pyridines. The authors already reported meta-selective introduction of trifluoromethyl and perfluoroalkyl groups and chlorine and bromine atoms into pyridines using the same strategy (Scheme 1c, ref. 52). In addition, meta- and para-selective introduction of alkyl groups into pyridine derivatives were also achieved by the same strategies by the same group (ref. 53). Therefore, this reaction can be considered as the extension of the authors' previous reports, and this manuscript is not suitable for publication in Nat. Commun. Before resubmission elsewhere, appropriate revisions and corrections are required according to the following comments;

1. Because difluoromethyl group is introduced stepwise, "difluoromethylation" is strange in the title. This word should be replaced with "formal difluoromethylation" or "introduction of difluoromethyl group".
2. In Scheme 2b, the other meta-position is also possible reaction site. Why the reaction does not occur at the other meta-position?
3. Is it possible to introduce difluoromethyl group at the meta-position of isoquinolines?
4. In almost all entries, the yields of the products are moderate. Why? Recovery of starting materials? Or formation of byproducts?
5. In the para-selective C-H difluoromethylation, is it necessary to promote the reaction via intermediate 6-H+/CSA- in Figure 2c? How about using standard N-alkenyl or N-alkylpyridinium salts?
6. In Figure 3, is it necessary to use pyridines with a (hetero)aryl group at the ortho-position or amide group at the meta-position?
7. The reaction schemes in Figures 4b and 4c are difficult to see, especially colored circles. Standard reaction schemes should be drawn.
8. In the title of Figure 1c, one of the same phrases should be deleted.

Reviewer #2 (Remarks to the Author):

Studer and co-workers proposed a novel method for switchable meta- and para-difluoromethylation of pyridines in this paper. Difluoromethyl molecules hold promise in medicinal chemistry, where regioisomers of difluoromethyl molecules are crucial in structure-activity relationship studies. This study successfully achieves selective CF₂H functionalization of pyridines by appropriate choice of the combination of the substrates and

the reagents (radical). The presented method also demonstrated proof-of-concept for late-stage functionalization, proving its high synthetic potential and applicability a diverse research fields including medicinal and chemical biology.

While the synthetic methodology is well-presented and easy to comprehend, I feel there is room for improvement regarding the impression that the choice of reagents and substrates merely combines known methods. To meet the high criteria for publication in Nature Communications, it would be necessary to include an academic discussion on the crucial challenges and novelty of this work featured by difluoromethylation.

Comments:

1) This method controlled the regioselectivity by changing the reaction conditions, where both the structure of the fluoroalkyl radical and the substrate were changed.

Wouldn't the structure of the substrate (intermediate) also influence selectivity?

Providing a comprehensive summary of reactivity based on combinations of radicals and substrates would enhance the clarity of this manuscript.

For this, the following references may be useful: (a) Nature Rev. Chem. 5, 486-499 (2021), (b) Chem. Rec. 23, e202300202 (2023), (c) Angew. Chem. Int. Ed. e202318377 (2024).

2) In Figure 2, the reaction of arylated pyridine with difluoromethyl radicals was depicted. What would be the outcome if substrate 3 were used instead?

Controlling the regioselectivity of the reaction of substrate 3, which lacks any substituent, would be particularly challenging.

Reviewer #3 (Remarks to the Author):

"Switchable meta- and para-C–H-Difluoromethylation of Pyridines"

Overview

Studer and coworkers apply their recently developed pyridine methodology to C–H difluoromethylation reactions that can be regiocontrolled between the meta and para positions depending on the reaction conditions and reagents employed. Their introduction nicely captures the importance of difluoromethylated products in medicinal and agrochemistry programs and accurately describes the current suite of chemical processes that can install this fluoroalkyl group. Indeed, there is still much room for improvement in the methodology available to practicing chemists.

The methods in this paper center on dearomatized oxazino pyridine intermediates that they have previously showed are valuable intermediates for pyridine functionalization with electrophiles and radical intermediates. Here, they found that meta selectivity occurs when they use a difluoromethyl surrogate reagent, and the reaction outcome is generally biased towards mono over difunctionalized products. They exploit acid to bias the reaction equilibrium to an N-substituted pyridinium salt to switch the reaction to the para-position of pyridines. The paper then focuses on the scope of the reaction, including building block-type pyridines, pyridine-containing drugs, and one-pot processes.

Critique

This paper is an excellent work that results in genuinely useful processes that are almost certain to gain traction in the pharmaceutical and agrochemical industries. There are only a small number of ways to access these products, and to my knowledge, this is the only

process that can reliably difluoromethylate the pyridine 5-position. The reaction protocols are relatively simple and easily accessible for practitioners. I was particularly impressed by the scope of each of the processes; the authors were careful to include a variety of substitution patterns and an array of functional groups. Late-stage functionalization looks viable, and the one-pot processes are appreciated from a convenience standpoint. Similarly, the sequential difunctionalization reactions are a useful addition. The paper is well-written and clearly explains all parts of reaction development and scope exploration. As such, I would strongly recommend the publication of this work as is, and some very minor points are mentioned below for the author's attention. Congratulations on an excellent piece of work.

Is TEMP not more commonly known as TMP?

Do the authors have any sense of what influences the mono to difunctionalization ratios? It looks like there might depend on the electron-donating capacity of the 4-position substituent, but not necessarily a clear cut phenomenon.

Response to Reviewer #1:

Comment 1: The authors report the introduction of difluoromethyl group into the meta- and para-positions of pyridine derivatives. The meta-selectivity is achieved by the radical addition of difluoromethyl radical to dearomatized pyridine derivatives by oxazino pyridine intermediates. On the other hand, the para-selectivity is realized by in situ transformation of the oxazino pyridines to pyridinium salts. The authors also investigated late-stage meta- and para-difluoromethylation of pyridine derivatives including drugs and double functionalization of pyridines. The authors already reported meta-selective introduction of trifluoromethyl and perfluoroalkyl groups and chlorine and bromine atoms into pyridines using the same strategy (Scheme 1c, ref. 52). In addition, meta- and para-selective introduction of alkyl groups into pyridine derivatives were also achieved by the same strategies by the same group (ref. 53). Therefore, this reaction can be considered as the extension of the authors' previous reports, and this manuscript is not suitable for publication in Nat. Commun. Before resubmission elsewhere, appropriate revisions and corrections are required according to the following comments;

Our response: We thank reviewer 1 for his/her suggestions to improve the quality of our manuscript. Given the importance of difluoromethylated pyridines, developing novel methods for the C-H difluoromethylation of pyridines is of high importance in our eyes. We agree that the strategy of using oxazino pyridines for regioselective pyridine C-H functionalization was reported by us before. However, its application towards the highly important difluoromethylation is unknown. Importantly, currently there exists no method that allows *meta*-C-H difluoromethylation of pyridines. Therefore, the focus of the current paper lies in the available product structures, and we believe that the two methods introduced should find use in pharmaceutical and agrochemical industry. Considering the *para*-difluoromethylation, the use of trifluoroacetic acid anhydride in combination with H₂O₂ for Minisci-type alkylation is unknown (that valuable reagent was used by Sodeoka for alkene difluoromethylation). We feel that this is of importance as the active reagent is readily prepared in situ at low temperature from cheap and commercial precursors and the active reagent also provides the oxidation equivalent necessary to complete a Minisci-alkylation. The low temperature is likely also of relevance to get high *para*-selectivity. A comment along those lines was made in the revised version of the manuscript.

Comment 2: Because difluoromethyl group is introduced stepwise, “difluoromethylation” is strange in the title. This word should be replaced with “formal difluoromethylation” or “introduction of difluoromethyl group”.

Our response: We agree with reviewer 1, the title was modified: ‘Introduction of the difluoromethyl group at the *meta*- or *para*-position of pyridines through regioselectivity switch’.

Comment 3: In Scheme 2b, the other meta-position is also possible reaction site. Why the reaction does not occur at the other meta-position?

Our response: We addressed that important point in the manuscript and added the following sentences: “The initial radical difluoromethylation always preferred to occur at the more reactive δ -position of the dienamine entity on the oxazino pyridine (**14-17**), as a more stabilized radical intermediate is formed through δ -addition, which is governed by a larger resonance stabilization. This high δ -regioselectivity was previously also observed for the radical *meta*-trifluoromethylation.⁵² In nearly all cases, the mono-functionalized product was formed with complete δ -regioselectivity. Only for oxazino pyridines that carry electron-rich aryl groups

on the α -position, the β -regioisomer was identified as a minor byproduct (see **14-16**). Considering double difluoromethylation, once the first CF_2COPh -group is installed, the remaining β -position is less nucleophilic, because of the electron-withdrawing effect exerted by the CF_2COPh -group. Accordingly, selective monodifluoromethylation was achieved in several cases (ratio $m:d > 20:1$, **9**, **12**, **13**, **15**, **17** and **26**). However, the β -position of the δ - CF_2COPh -functionalized oxazino pyridine remains reactive depending on the additional oxazino pyridine substituents. Considering γ -arylated oxazino pyridines as substrates, mono/di-functionalization selectivity decreases as a function of the electron-donating ability of the para-substituent on the aryl group, in line with our hypothesis (see **8**, **9** and **10**). Further, for α - and γ -substituted oxazino pyridines, steric effects will likely also slow down the second C–H-functionalization.”

Comment 4: Is it possible to introduce difluoromethyl group at the meta-position of isoquinolines?

Our response: We have tested the feasibility of difluoromethylation of the isoquinoline. Unfortunately, only a trace amount of the desired product was observed. This result was included in the Supporting Information (see page 24). The following comment regarding the failed experiment was added to the revised manuscript: “Isoquinolines could not be difluoromethylated through this strategy.”

Comment 5: In almost all entries, the yields of the products are moderate. Why? Recovery of starting materials? Or formation of byproducts?

Our response: The moderate yields are mainly caused by the following three reasons (see Figure below).

- A two-fold extraction was conducted, and some material got lost during this multi-step operation.
- Most difluoromethyl pyridines synthesized in this work are volatile. Accordingly, part of the product gets lost during solvent removal (by rota-vap) and during drying on vacuum pump.
- Because the acidity of silica-gel, the basic pyridines are partly lost during purification.
- In the case of meta-substituted pyridines, the corresponding oxazino pyridines are formed as diastereoisomeric mixtures of the two regioisomers. The minor regioisomer was found to be unreactive, and the starting pyridine derived from that regioisomer could be recovered after hydrolysis.

These explanations and also the corresponding figure can be found in the revised Supporting Information on page 24. A comment in the manuscript alluding to Page 24 of the SI was added to the revised version.

Comment 6: In the *para*-selective C-H difluoromethylation, is it necessary to promote the reaction via intermediate 6-H⁺/CSA- in Figure 2c? How about using standard *N*-alkenyl or *N*-alkylpyridinium salts?

Our response: The use of *N*-alkenyl or *N*-alkylpyridinium salts for *para*-selective Minisci-alkylation was recently investigated by Baran et al. (*J. Am. Chem. Soc.* **2021**, *143*, 11927). The tested *N*-alkenyl pyridinium salts did not engage in the Minisci-alkylation, while alpha-branched *N*-alkylpyridinium salts gave the *para*-alkylated products with high regioselectivity. Although Baran did not address the difluoromethylation, it is likely that as for other alkyl radicals a high *para*-selectivity can be obtained for his system using “our” difluoromethylation conditions. However, selective *para*-functionalization of more complex pyridines was not documented in the Baran paper and a regioselectivity switch is not possible using their strategy.

Comment 7: In Figure 3, is it necessary to use pyridines with a (hetero)aryl group at the *ortho*-position or amide group at the *meta*-position?

Our response: It is not necessary for pyridines to have a (hetero)aryl group at the *ortho*-position or an amide group at the *meta*-position, as products **5**, **8-13**, **37** (without any substituents at the *ortho*-position) and **29**, **52** (with an alkynyl group at the *ortho*-position) could be prepared. Difluoromethylated pyridines with substituents other than an amide group at the *meta*-position could be prepared as well (see **18**, **19**, **21-25**).

Comment 8: The reaction schemes in Figures 4b and 4c are difficult to see, especially colored circles. Standard reaction schemes should be drawn.

Our response: We thank reviewer 1 for this kind suggestion. Figures 4b and 4c have been improved.

Comment 9: In the title of Figure 1c, one of the same phrases should be deleted.

Our response: The figure heading was modified accordingly.

Response to Reviewer #2:

Comment 1: Studer and co-workers proposed a novel method for switchable *meta*- and *para*-difluoromethylation of pyridines in this paper. Difluoromethyl molecules hold promise in medicinal chemistry, where regioisomers of difluoromethyl molecules are crucial in structure-activity relationship

studies. This study successfully achieves selective CF₂H functionalization of pyridines by appropriate choice of the combination of the substrates and the reagents (radical). The presented method also demonstrated proof-of-concept for late-stage functionalization, proving its high synthetic potential and applicability a diverse research fields including medicinal and chemical biology.

While the synthetic methodology is well-presented and easy to comprehend, I feel there is room for improvement regarding the impression that the choice of reagents and substrates merely combines known methods. To meet the high criteria for publication in Nature Communications, it would be necessary to include an academic discussion on the crucial challenges and novelty of this work featured by difluoromethylation.

Our response: We thank reviewer 2 for his/her kind recommendation. We added discussion regarding the regioselectivity problem of the difluoromethylation (see also comment to referee 1). In addition, we also discussed in more detail the problem of mono versus bis-trifluoromethylation of the oxazino pyridines. These critical points further highlight some of the challenges associated with this approach. Moreover, we also commented on the challenges associated with the isolation and purification of difluoromethylated pyridines (comment in the paper, and a new Figure was added to the SI on page 24). As far as we know, we covered well the literature on the para-difluoromethylation and also highlighted that the meta-difluoromethylation is unknown (see Figure 1b). The facts that meta-difluoromethylated pyridines occur in drugs (see Figure 1a) and that no direct method for their preparation through C-H functionalization exists, indicates the challenge associated with their direct preparation. In the text we highlighted that meta,para-switchable regioselective difluoromethylation is unknown. As far as we are aware, difluoroacetic anhydride in combination with H₂O₂, as introduced by Sodeoka, has never been used in the Minisci-alkylation. A sentence along those lines was added “To our knowledge, this readily available radical difluoromethylation reagent studied by Sodeoka for alkene difluoromethylation⁵⁸ has not yet been used in Minisci-type alkylation reactions.” to further underline the novelty of the work.

Moreover, in the introduction of our paper on page 2 we alluded to the intrinsic problems (challenges) of the pyridine C-H functionalization: “Known methods for C–H-functionalization of pyridines are largely restricted to the ortho- and para-positions due to the electronic nature of the pyridine core.^{17-21,32} ...” (the following sentences of the intro discuss the current solutions to that challenge.

Comment 2: This method controlled the regioselectivity by changing the reaction conditions, where both the structure of the fluoroalkyl radical and the substrate were changed.

Wouldn't the structure of the substrate (intermediate) also influence selectivity?

Providing a comprehensive summary of reactivity based on combinations of radicals and substrates would enhance the clarity of this manuscript.

For this, the following references may be useful: (a) Nature Rev. Chem. 5, 486-499 (2021), (b) Chem. Rec. 23, e202300202 (2023), (c) Angew. Chem. Int. Ed. e202318377 (2024).

Our response: The regioselectivity mainly depends on the substrate (oxazino pyridine intermediate). In the oxazino pyridine, the meta-positions are activated as nucleophilic sites, and therefore react well with electrophilic radicals. In the pyridinium salt, the ortho- and para-positions are both electronically activated. While the ortho positions are well blocked by the *N*-substituent, reaction with the C-radical occurs with high selectivity at the para-position. Moreover, the low temperature used for the para-difluoromethylation further helps in getting a high *para*-selectivity.

Considering the *meta*-functionalization, substituents in the oxazino pyridines influence reactivity. Regioselectivity problems and also difunctionalization may occur depending on the nature of the substituent. This substituent effect is discussed in the revised version (see also comment to referee 1 above).

Considering the attacking C-radical, polyfluoroalkyl radicals are typically electrophilic in nature, but can show nucleophilic character (*Nature Rev. Chem.* **2021**, 5, 486). While $\bullet\text{CF}_2\text{H}$ is not electrophilic enough to add to the oxazino pyridine, the $\bullet\text{CF}_2(\text{COPh})$ was to realize the *meta*-functionalization, as the electron-withdrawing nature of benzoyl group enhances the electrophilicity of that difluoroalkyl radical (*Angew. Chem. Int. Ed.* **2024**, e202318377). The higher electrophilicity of this masked difluoromethyl radical was mentioned in the manuscript. Meanwhile, $\bullet\text{CF}_2\text{H}$ shows nucleophilic character in functionalization of electron-deficient arenes (*J. Am. Chem. Soc.* **2012**, 134, 1494; *Chem. Rec.* **2023**, 23, e202300202), and accordingly is able to react with the pyridinium salts. The aforementioned references have been added to the revised manuscript.

Comment 3: In Figure 2, the reaction of arylated pyridine with difluoromethyl radicals was depicted. What would be the outcome if substrate 3 were used instead?

Controlling the regioselectivity of the reaction of substrate 3, which lacks any substituent, would be particularly challenging.

Our response: Because the 4-(difluoromethyl)pyridine **37** is highly volatile, initially we did not use oxazino pyridine **3** as a substrate for the *para*-difluoromethylation. That requested experiment was conducted in the revision and the corresponding result is presented in Figure 3 of the revised manuscript. The following sentence was added to the revised manuscript: "The 4-(difluoromethyl)pyridine **37** lacking any additional substituent was obtained in 61% along with 15% of the corresponding *ortho,para*-bistrifluoromethylated pyridine. Of note, *ortho*-difluoromethylation was not observed for all other cases, showing the very good *ortho*-shielding effect of the N-substituent in these pyridinium salts."

Response to Reviewer #3:

Comment 1: "Switchable meta- and para-C–H-Difluoromethylation of Pyridines"

Overview

Studer and coworkers apply their recently developed pyridine methodology to C–H difluoromethylation reactions that can be regiocontrolled between the meta and para positions depending on the reaction conditions and reagents employed. Their introduction nicely captures the importance of difluoromethylated products in medicinal and agrochemistry programs and accurately describes the current suite of chemical processes that can install this fluoroalkyl group. Indeed, there is still much room for improvement in the methodology available to practicing chemists.

The methods in this paper center on dearomatized oxazino pyridine intermediates that they have previously showed are valuable intermediates for pyridine functionalization with electrophiles and radical intermediates. Here, they found that meta selectivity occurs when they use a difluoromethyl surrogate reagent, and the reaction outcome is generally biased towards mono over difunctionalized products. They exploit acid to bias the reaction equilibrium to an N-substituted pyridinium salt to switch the reaction to the para-position of pyridines. The paper then focuses on the scope of the reaction, including building block-type pyridines, pyridine-containing drugs, and one-pot processes.

Critique

This paper is an excellent work that results in genuinely useful processes that are almost certain to gain traction in the pharmaceutical and agrochemical industries. There are only a small number of ways to access these products, and to my knowledge, this is the only process that can reliably difluoromethylate the pyridine 5-position. The reaction protocols are relatively simple and easily accessible for practitioners. I was particularly impressed by the scope of each of the processes; the authors were careful to include a variety of substitution patterns and an array of functional groups. Late-stage functionalization looks viable, and the one-pot processes are appreciated from a convenience standpoint. Similarly, the sequential difunctionalization reactions are a useful addition. The paper is well-written and clearly explains all parts of reaction development and scope exploration. As such, I would strongly recommend the publication of this work as is, and some very minor points are mentioned below for the author's attention. Congratulations on an excellent piece of work.

Our response: We really appreciate the reviewer's supportive comments.

Comment 2: Is TEMP not more commonly known as TMP?

Our response: We thank reviewer 3 for alluding to this problem, 'TEMP' has been changed to 'TMP' throughout the manuscript.

Comment 3: Do the authors have any sense of what influences the mono to difunctionalization ratios? It looks like there might depend on the electron-donating capacity of the 4-position substituent, but not necessarily a clear cut phenomenon.

Our response: Considering γ -arylated oxazino pyridines (4-position substituent) as substrates, mono/difunctionalization selectivity decreases as a function of the electron-donating ability of the *para*-substituent on the aryl group (see for example products **8**, **9** and **10**). This is in line with our hypothesis/discussion that electron-donating substituents enhance the reactivity for both mono- as well as for the difunctionalization. Further, for α - and γ -substituted oxazino pyridines, steric effects will likely also slow down the second C-H functionalization. The excellent mono-functionalization selectivity observed for the γ -phenoxy oxazino pyridine is currently not understood (see product **13** in the manuscript), as one would expect a large amount of bis-difluoromethylation due to the electron-donating effect exerted by the phenoxy group.

The initial radical difluoromethylation occurs at the more reactive δ -position of the dienamine entity on the oxazino pyridine (see products **14-17**), as a more stabilized radical intermediate is formed through δ -addition, which is governed by a larger resonance stabilization. In nearly all cases, the mono-functionalized product was formed with complete δ -regioselectivity. Only for oxazino pyridines that carry electron-rich aryl groups on the α -position, the β -regioisomer was identified as a minor byproduct

(see products **14-16**), while the *ortho*-(2,4-difluorophenyl) oxazino pyridine provided exclusively the mono-functionalization product (see product **17**). The *ortho*-(thien-2-yl)-substituted oxazino pyridine (see below) showed very high β -selectivity with the β -isomer formed as the major regioisomer (an exception). We made a similar observation also for the trifluoromethylation. Thus, it seems that an electron-rich *ortho*-(hetero)aryl substituent increases β -reactivity. Unfortunately, we could not get the thienyl-product in hand and therefore this example could not be presented in the manuscript.

Considering double difluoromethylation, once the first CF_2COPh -group is installed, the remaining β -position is less nucleophilic, because of the electron-withdrawing effect exerted by the CF_2COPh -group. Accordingly, selective mono-difluoromethylation was achieved in several cases (ratio $m:d > 20:1$, **9**, **12**, **13**, **15**, **17** and **26**). However, the β -position of the δ - CF_2COPh -functionalized oxazino pyridine remains reactive depending on the additional oxazino pyridine substituents. This discussion is now provided in the revised manuscript.

REVIEWERS' COMMENTS

Reviewer #1 (Remarks to the Author):

This reviewer reviewed the former version of this manuscript. Judging from the results of regioselective introduction of difluoromethyl groups at the meta- and para-positions of pyridines, this research is novel and excellent. However, as this reviewer commented in the former review, the authors have already reported on the reaction design used in this study to switch the meta- and para-positions of pyridines. Therefore, the lack of examples of regioselective introduction of difluoromethyl groups is hardly sufficient for acceptance in top-level journals. Depending on one's point of view, it could be seen as simply expanding the range of the reagents or, as reviewer 2 stated, as simply combining known methods. However, in this revision, the authors have addressed the points raised by the three reviewers. If Reviewer 2 deems this revised version suitable for publication, it will be considered suitable for publication, as it is the novel example of regioselective introduction of difluoromethyl groups at the meta- and para-positions of pyridines.

Reviewer #2 (Remarks to the Author):

The authors have appropriately addressed the points I raised and have improved the manuscript, resulting in a better-quality paper. Therefore, I recommend the publication of this manuscript in Nature Communications.

Reviewer #3 (Remarks to the Author):

"Switchable meta- and para-C–H-Difluoromethylation of Pyridines" Resubmission

I have carefully examined the responses from the author to my questions and the questions from the other reviewers. They have taken each point very seriously and provided thorough and reasoned responses in my viewpoint. I am happy with the revised document and changes to the supporting information. I look forward to seeing this work in print.